# Thermal Analysis Based on Dynamic Performance of Rocker Arm Full-Type Needle Bearings

**Guangtao Zhang, Bing Su \*, Fengbo Liu, Wenhu Zhang and Haisheng Yang**

School of Mechatronics Engineering, Henan University of Science and Technology, Luoyang 471003, China; hkd_zhanggt@163.com (G.Z.); hnly_lfb@163.com (F.L.); 9905721@haust.edu.cn (W.Z.); yanghaisheng1975@163.com (H.Y.)
\* Correspondence: subing@haust.edu.cn; Tel.: +86-0379-64231479

**Abstract:** Based on a dynamic analysis of rolling bearings, the equations for rocker arm full-type needle bearings were established by considering the traction coefficients of FVA-M reference lubricating oil, and then they were solved by the GSTIFF (Gear Stiff) integer algorithm with variable steps. The influence of working conditions on friction power consumption and the lubricant's convective coefficients were investigated. Then, on the basis of the heat generation and heat transfer mechanisms, the frictional power consumption was used as the boundary condition of the bearings' simulation model. Finally, temperature fields were calculated by the finite element method. The results showed that the overall value of frictional power consumption increased gradually with the increase in either the radial load or the rotation speed. The presence or absence of lubricating oil film in the contact area affected the heat conduction of the bearing, resulting in a temperature difference. Compared with the temperature of the radial load exerted on the bearing, the maximum temperature was more sensitive to the variations in the rotation speed. When running under the conditions of a fatigue life test, the steady-state temperature value of the bearing gradually decreased from the outer raceway to the needle roller and the outer ring surface, and then to the central shaft. The maximum temperature rise was 25.9 °C relative to the ambient temperature.

**Keywords:** full-type needle bearing; dynamics performance; temperature fields; lubricating oil

## 1. Introduction

A rocker arm full-type needle bearing is used between a cam and a rocker to control the reciprocating movement of a valve lever in completing the intake and exhaust functions [1]. Due to the complex structure of the valve train and the flexible and changeable positions of the components, it is necessary to analyze the dynamics and temperature field of the valve train under different loads and speeds [2,3]. Dynamic performance analysis can determine the stress and friction torque of the main contact parts of the bearing, thereby determining the heat source value that can affect the bearing's performance. The changes in bearing temperature caused by frictional heat will cause the thermal deformation of components, greatly affecting the internal structural dimensions of the bearing [4]. A reduction in the working clearance can lead to damage on the surface of the rolling bearing and, in severe cases, shorten the life of the bearing. Those phenomena will affect the running accuracy of the bearing and cause varying degrees of harm to the engine.

Thermal analysis of rolling bearings is the study of the heat generated by friction between bearing components and lubrication. The friction properties of the lubricants affect the tribological behavior of the rolling bearings [5,6]. Palmgren [7] carried out friction tests on bearings of various types and sizes, and regressed the test data to obtain an empirical formula for calculating bearing friction torque. Ma et al. [8] classified and discussed the existing experimental research regarding heat generation and heat transfer of bearing components, and briefly described the calculation method of combining theoretical models to predict the thermal problems of high-speed bearings. Hatazawa et al. [9] established a

mathematical model of the friction torque of thrust needle roller bearings and studied the influence of working conditions and structural parameters on the dynamic performance of thrust needle roller bearings, but that study did not mention the thermal analysis of the bearings. Li et al. [10] used ADAMS software to analyze the dynamics of cylindrical roller bearings and determined the friction and positive pressure between the rolling elements and the inner raceway at different positions. Then, they analyzed the contact stress between rollers and the inner raceway with ANSYS Workbench. Based on the dynamic method, the friction heat inside the bearing was obtained. On the basis of this information, the temperature field of the bearing was analyzed. Wang et al. [11,12] established an analysis model based on the quasi-static method to calculate the heat source of high-speed cylindrical roller bearings and studied the effects of speed, load, lubricating oil, and working temperature on heat generation. Subramaniam et al. [13] used the Finite Element Method to analyze heat transfer in a ceramic conventional ball bearing and study the heat dissipation, temperature profile, and thermal stresses occurring in a bearing as a function of rotational speed. Liming et al. [14] used the finite element method to calculate the temperature field of the sliding block under different loads and speeds, and verified the feasibility of the finite element method in analyzing the temperature field.

For the finite element method to solve the bearing temperature field problem, the accurate determination of the boundary conditions determines the accuracy of the bearing thermal analysis. In the process of determining the boundary conditions, most of the current literature assumes that the bearing's rolling elements have the same heat transfer form during the rotation of the bearing. The above review of the relevant literature indicated that the internal temperature of the bearing was generally evenly distributed, and the calculation of the bearing was based on the heat in the stable operation stage. The research results were obtained by using the finite element method to deal with such boundary conditions and were verified by experiments; the results were comparable to previous experimental data, such as the data provided in [15,16].

Many previous studies focused on the calculation of the friction torque based on the lubrication and the load effects to determine the heat source of the needle roller bearing. However, the variations in the elasto-hydrodynamic traction coefficient affect the interaction force of the bearing's elements. Based on dynamic analysis, the programming language FORTRAN was used in this paper to compile custom functions to realize the analysis of the force balance of various components in the bearing. The heat of bearings lubricated by FVA-M lubricating oil was calculated. Considering the heat transfer coefficient of the lubricating oil film in the contact area, steady-state thermal analysis was used to determine the internal equilibrium temperature field of the bearing.

## 2. Dynamic Model of a Rocker Needle Roller Bearing

The research object was a certain type of rocker arm full-type needle bearing, which is used for the rocker arm valve mechanism and mainly includes an outer ring in rolling contact with the engine camshaft. A central shaft is fixed on the rocker arm, and multiple rolling elements are arranged between the outer ring and the central shaft. Together, these components form the rocker bearing.

When the bearing is working, the central shaft is fixed. Therefore, it is assumed that the central shaft is connected with the ground in the dynamic analysis, and only the outer ring rotates together with the action of the cam. The axis of rotation is along the X direction. The center of mass of each component coincides with the center of gravity, and the surface of the component is an ideal surface. In order to establish a dynamic model of the bearing, the following three coordinate systems, shown in in Figure 1, were defined: $\{O; X, Y, Z\}$ is a fixed coordinate system; $\{o_o; x_o, y_o, z_o\}$ is a coordinate system of the outer ring; and $\{o_{rj}; x_{rj}, y_{rj}, z_{rj}\}$ is the center of the mass coordinates of the needles.

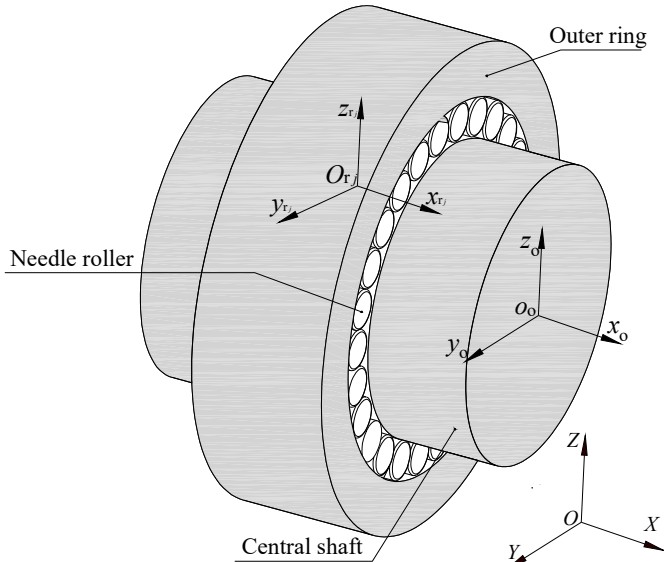

**Figure 1.** Coordinate systems.

Based on the following assumptions, Newton's law of motion and classical Euler dynamic equations can be used to derive the force and motion of the components of the rocker needle roller bearing, as follows:

1.  The contact deformation between the needle roller and the raceway conforms to Hertzian elliptical contact theory, and the contact state is full line contact.
2.  The influences of the frictional moment caused by the skew and tilt of the needle in motion are not considered.
3.  The needle rollers are evenly distributed in the initial state of simulation, with a certain circumferential clearance.

### 2.1. Motion Analysis of Needle Rollers

The movement relationship among the components is shown in Figure 2. The central shaft is fixed. Suppose the rotation angular velocity of the outer ring is $\Omega_o$, and the needle rollers' revolution angular velocity is $\Omega_m$, rotating at a uniform speed. If the needle rollers are fixed, the movement of the bearing assembly can be equivalent to the angular velocity of the outer ring $(\Omega_o - \Omega_m)$, and the angular velocity of the central shaft relative to the needle roller is $\Omega_m$. Therefore, the velocities of the contact area can be obtained, which are $\vec{V}_{rj}$, $\vec{V}_i$, $\vec{V}_o$, respectively.

$$\vec{V}_{rj} = \frac{1}{2}D_w\vec{\omega}_{rj} \tag{1}$$

$$\vec{V}_i = \frac{1}{2}\left(D_{wp} - D_w\right)\vec{\Omega}_m \tag{2}$$

$$\vec{V}_o = \frac{1}{2}\left(D_{wp} + D_w\right)\left(\vec{\Omega}_o - \vec{\Omega}_m\right) \tag{3}$$

The relative sliding speeds of the contact points between the central shaft, the outer ring raceway, and the needle rollers and adjacent rollers are shown as Equations (4)–(6), respectively.

$$\vec{V}_{ij} = \vec{V}_i - \vec{V}_{rj} \tag{4}$$

$$\vec{V}_{oj} = \vec{V}_o - \vec{V}_{rj} \tag{5}$$

$$\Delta\vec{V}_{rj} = \vec{V}_{rj} - \vec{V}_{r(j+1)} \tag{6}$$

where the subscript 'i' represents the central shaft; 'o' represents the outer ring; 'r' represents the needle roller; '*j*' represents the needle number; $\omega$ is the angular velocity of the needle roller; $D_{\text{wp}}$ is the center diameter of the needle roller group; and $D_{\text{w}}$ is the needle diameter.

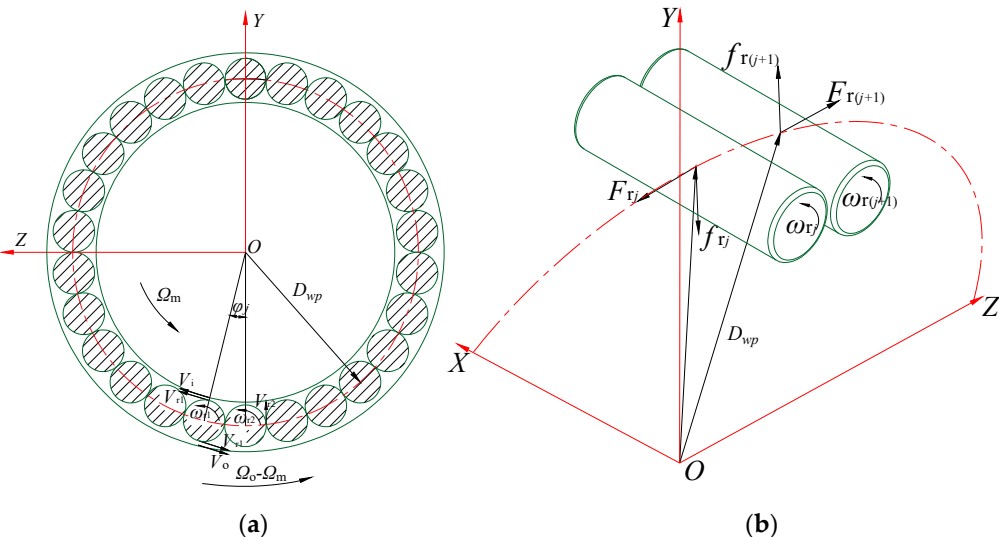

**Figure 2.** Motion and load status of full-type needle bearing: (**a**)Motion status; (**b**) Load status.

### 2.2. Force Analysis of Needle Roller

As shown in Figure 3, $N_j^{\text{i}}$ and $N_j^{\text{o}}$ are the normal contact forces between the *j*th needle and the raceway. $T_j^{\text{i}}$ and $T_j^{\text{o}}$ are the traction forces between the *j*th needle and the raceway. $M_{\text{N}j}^{\text{i}}$ and $M_{\text{N}j}^{\text{o}}$ are the moments caused by the normal contact force. $M_{\text{T}j}^{\text{i}}$ and $M_{\text{T}j}^{\text{o}}$ are the moments caused by traction forces between the *j*th needle and the raceway. $F_{\text{r}j}$ is the centrifugal force of the *j*th needle roller; $q_{jm}^{\text{i}}$, $q_{jm}^{\text{o}}$ are the normal contact forces between the *m*th needle roller and the raceway. $T_{jm}^{\text{i}}$, $T_{jm}^{\text{o}}$ are the traction forces between the *m*th needle roller and the raceway. *n* is the number of slices of the needle roller. The calculation of the forces and moments can then be carried out [17].

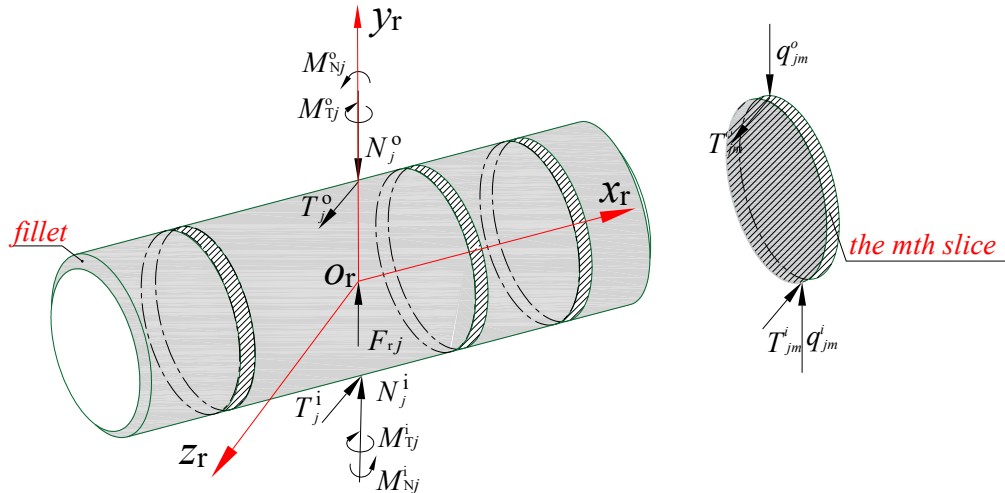

**Figure 3.** Schematic diagram of needle forces.

On the basis of the dynamic relationship, the nonlinear equation of the $j$th needle roller can be written as follows:

$$\begin{cases} m_r \ddot{y}_{rj} = N_j^o \cos \varphi_j - N_j^i \cos \varphi_j - F_{rj} \cos \varphi_j \\ \qquad + T_j^i \sin \varphi_j - T_j^o \sin \varphi_j \\ m_r \ddot{z}_{rj} = -N_j^o \sin \varphi_j + N_j^i \sin \varphi_j + F_{rj} \sin \varphi_j \\ \qquad + T_j^i \cos \varphi_j - T_j^o \cos \varphi_j \\ J_{rx} \dot{\omega}_{rx} = T_j^i \frac{Dw}{2} + T_j^o \frac{Dw}{2} \\ J_{ry} \dot{\omega}_{ry} = M_{Tj}^i \cos \varphi_j + M_{Tj}^o \cos \varphi_j - M_{Nj}^i \sin \varphi_j \\ \qquad - M_{Nj}^o \sin \varphi_j \\ J_{rz} \dot{\omega}_{rz} = -M_{Tj}^i \sin \varphi_j - M_{Tj}^o \sin \varphi_j - M_{Nj}^i \cos \varphi_j \\ \qquad - M_{Nj}^o \cos \varphi_j \end{cases} \qquad (7)$$

where $m_r$ is the mass of the needle; $\ddot{y}_{rj}$ and $\ddot{z}_{rj}$ are the displacement accelerations of mass of the $j$th needle mass center in {O; X, Y, Z} ; $J_{rx}$, $J_{ry}$, and $J_{rz}$ are the moments of inertia under the needle in {O; X, Y, Z}; and $\dot{\omega}_{rx}$, $\dot{\omega}_{ry}$, and $\dot{\omega}_{rz}$ are the angular accelerations of the needle in {O; X, Y, Z}.

The interaction model between two adjacent rollers is shown in Figure 2. According to the theory of contact and collision between rigid bodies, the expression of the mathematical model of the force of adjacent rollers is obtained as follows:

$$F_{rj(j+1)} = \left\{ \begin{array}{ll} 0 & (q \leq q_1) \\ K_n(q_1 - q) - c_{max} \dot{q} step(q, q_1 - d, 1, q_1, 0) & (q > q_1) \end{array} \right\} \qquad (8)$$

where $q$ is the distance variable; $q_1$ is the relative speed; $\dot{q}$ is the derivative of the relative speed; $e$ is the exponent of the force; $K_n$ is the stiffness coefficient; $c_{max}$ is the maximum damping coefficient; $d$ is the cut-in amount when the damping reaches the maximum value; and *step* (*) is the step function.

Therefore, the friction force $f$ between adjacent needle rollers can be obtained as follows:

$$f_{rj} = \mu F_{rj} \qquad (9)$$

$$f_{r(j+1)} = -\mu F_{r(j+1)} \qquad (10)$$

The traction coefficient $\mu$, relating to the sliding speed between the needle rollers, can be determined according to the given traction model.

### 2.3. Nonlinear Equation of the Outer Ring

Similarly, the dynamic equation of the outer ring can be obtained via the following equations from the relationship between motion and force:

$$\begin{cases} m_o \ddot{y}_o = \sum_{j=1}^{RN} \left( -N_j^o \cos \varphi_j + T_j^o \sin \varphi_j \right) - F_r \\ m_o \ddot{z}_o = \sum_{j=1}^{RN} \left( N_j^o \sin \varphi_j + T_j^o \cos \varphi_j \right) \\ J_{ox} \dot{\omega}_{ox} = \sum_{j=1}^{RN} T_j^o \frac{Dw}{2} \\ J_{oy} \dot{\omega}_{oy} = \sum_{j=1}^{RN} \left( M_{Tj}^o \cos \varphi_j - M_{Nj}^o \sin \varphi_j \right) \\ J_{oz} \dot{\omega}_{oz} = \sum_{j=1}^{RN} \left( -M_{Tj}^o \sin \varphi_j - M_{Nj}^o \cos \varphi_j \right) \end{cases} \qquad (11)$$

where $m_o$ is the mass of the outer ring; $\ddot{y}_o$ and $\ddot{z}_o$ are the displacement accelerations of the center of mass of the outer ring in inertial coordinate system {O;X,Y,Z} ; $J_{ox}$, $J_{oy}$, and $J_{oz}$ are

the moments of the outer ring in inertial coordinate system {O;X,Y,Z}, $\dot{\Omega}_{ox}$, $\dot{\Omega}_{oy}$ and $\dot{\Omega}_{oz}$ are the angular accelerations of the outer ring in inertial coordinate system {O;X,Y,Z}; $F_r$ is the radial load; and $RN$ is the number of needle rollers.

In the above equations, the traction coefficient $\mu_{jm}$ of the oil is a variable value, which is obtained through experiments. The calculation formula of the oil traction coefficient is fitted from the test data according to the Gupta's four-parameter model. The FVA-M lubricant was provided by German Schaeffler Company; its traction coefficient was tested using the lubricant traction characteristic test ring, and can be expressed in the following equation [18]:

$$\mu_{jm}= (A + BS)e^{-CS_{jm}} + D \tag{12}$$

$$\begin{cases} A = -0.5164\overline{W}^{0.7102}\overline{U}^{0.5353}\overline{T}^{-0.4159} \\ B = 4.6747 \times 10^4\overline{W}^{-0.0725}\overline{U}^{0.2831}\overline{T}^{0.079635} \\ C = 3.4129 \times 10^{-6}\overline{W}^{-0.021}\overline{U}^{0.274}\overline{T}^{-0.4796} \\ D = 0.5164\overline{W}^{0.7102}\overline{U}^{0.5353}\overline{T}^{-0.4159} \end{cases} \tag{13}$$

where $\overline{W}$ is the dimensionless load parameter; $\overline{U}$ is the dimensionless speed parameter; $\overline{T}$ is the dimensionless temperature parameter; and $S_{jm}$ is the sliding-rolling ratio (the ratio of the sliding velocity to the rolling velocity) at the contact point between the $m$th slice of the $j$th needle and the inner or outer raceways.

## 3. Heat Generation and Heat Transfer Calculation Model for Rocker Needle Roller Bearings

### 3.1. Calculation Model of Heat Generation for Rocker Needle Roller Bearings

Due to a certain friction phenomenon among the rolling elements, the raceways, and the lubricating film, the bearing friction is not constant. According to Palmgren, the total frictional power consumption of a bearing should be determined by the frictional power between the rolling element and the ring, caused by the load and the lubricant's viscous frictional power consumption when the bearing is at a medium speed and subjected to a medium external load. Harris [19] modified the overall calculation of bearing friction power consumption through experiments based on Palmgren's work. He proposed a calculation method for local heat generation that takes into account the six major factors of rolling bearing friction. The dynamic parameters of the bearing can be analyzed by using a computer program, and then the local frictional heat calculation formula of the bearing element can be solved simultaneously.

Based on the local heat generation model proposed by Harris, the total friction power consumption of a rocker arm full-type needle bearing is comprised of the friction power consumption caused by the elastic hysteresis between the needle rollers and the raceways, the friction power consumption caused by the sliding friction between the needle rollers and the raceways, the friction power consumption caused by the viscous resistance between the oil and the bearing components, and the friction power consumption caused by the sliding between adjacent needle rollers.

#### 3.1.1. Frictional Power Consumption Caused by the Elastic Hysteresis between the Needle Rollers and the Raceways

When the needle roller rolls on the central shaft and the outer ring raceway, the frictional power consumption due to the elastic hysteresis of the material can be calculated as follows:

$$M_E = \sum_{j=1}^{RN} \sum_{m=1}^{n} \left(\xi\sqrt{\frac{\pi q_{jm}^{i(o)}}{2\eta D_w}}\delta_{jm}^{i(o)}\right) \tag{14}$$

$$E_R = M_E\left|\omega_{i(o)} - \omega_m\right| \tag{15}$$

where $\xi$ is the elastic hysteresis coefficient of the material, which is 0.01 preferable for steel; $\eta$ is the comprehensive elastic constant of the two contact bodies; and $\delta_{jm}^{i(o)}$ is the elastic deformation between the $m$th slice of the $j$th needle roller and rings.

### 3.1.2. Friction Power Consumption Caused by Sliding Friction between the Needle Rollers and the Rings

When the needle roller rolls on the central shaft and the outer ring raceway, the frictional power consumption caused by relative sliding can be calculated as follows:

$$E_{\mathrm{D}} = \sum_{j=1}^{RN} \sum_{m=1}^{n} \left( \mu_{jm}^{i(o)} q_{jm}^{i(o)} \Delta v_{jm}^{i(o)} \right) \tag{16}$$

where $\mu_{jm}^{i}$ and $\mu_{jm}^{o}$ are, respectively, the traction coefficients of the oil film between the $m$th slice of the $j$th needle and the inner and outer raceways, which are obtained by interpolation from the relationship between the slip-roll ratio and the elasto-hydrodynamic tration coefficient; and $\Delta v_{jm}^{i}$ and $\Delta v_{jm}^{o}$ are, respectively, the relative sliding speeds between the $m$th slice of the $j$th needle and the inner and outer raceways.

### 3.1.3. Friction Power Consumption Caused by Viscous Resistance

Due to the stirring effect of the lubricating oil during the rotation of the needle, the frictional power consumption caused by the lubricating oil film formed at the contact between the rolling elements and the inner and outer raceways is calculated as follows [17]:

$$E_{\mathrm{oil}} = \int_{0}^{2\pi} \frac{1}{8} C_{\mathrm{d}} \rho_m D_{\mathrm{w}} l (D_{\mathrm{wp}} \omega_{\mathrm{r}})^2 \mathrm{d}\varphi \tag{17}$$

where $C_{\mathrm{d}}$ is the flow resistance coefficient; $\rho_m$ is the density of oil and gas mixture; $l$ is the length of needle roller; and $\varphi$ is the position angle of the needle roller.

### 3.1.4. Sliding Friction Power Consumption between Adjacent Needle Rollers

The frictional power consumption due to the contact and collision between adjacent rollers can be expressed as follows:

$$E_{\mathrm{r}} = \sum_{j=1}^{RN} \mu F_{\mathrm{r}j} \left| \Delta \vec{V_{\mathrm{r}}} \right| \tag{18}$$

The total frictional power consumption can be expressed as follows:

$$E = E_{\mathrm{R}} + E_{\mathrm{D}} + E_{\mathrm{oil}} + E_{\mathrm{r}} \tag{19}$$

### 3.2. Heat Transfer Calculation Model for Rocker Needle Roller Bearings

For a bearing system, heat convection mainly includes the convection between the lubricant and the surface of the inner or outer raceway and the convection between the lubricant and the surface of the needle rollers. When the lubricant is oil, the performance of the lubricating oil film has a greater impact on improving the working conditions and the service life of the bearing; the convection of the lubricating oil film in the contact area should also be considered. The convection coefficient can be approximately calculated by Equations (20)–(21).

### 3.2.1. Convective Coefficient between Lubricant and Bearing Element Surface

$$\alpha_1 = 0.332 \frac{k}{D_{\mathrm{x}}} P_{\mathrm{r}}^{\frac{1}{3}} \left( \frac{V_{\mathrm{x}}}{v_0} \right)^{\frac{1}{2}} \tag{20}$$

where $k$ is the thermal conductivity of the lubricant; $P_r$ is the Prandtl number of the lubricant; $v_0$ is the kinematic viscosity of the lubricating oil; $D_{\mathrm{x}}$ is the radius of rotation, the outer ring taking the groove bottom diameter $D_{\mathrm{o}}$, the central axis taking the groove bottom

diameter $D_i$, and the roller needle taking $D_{wp}$; $V_x$ is the velocity of bearing component, $V_x = \frac{\pi}{60}n_r D_x$; $n_r = \frac{1}{2}(n_i(1 - \frac{D_w}{D_{wp}}) + n_o(1 + \frac{D_w}{D_{wp}}))$; $n_r$ is the revolution speed of the needle roller; and $n_i$ and $n_o$ are the speeds of the central shaft and the outer ring, respectively.

### 3.2.2. Thermal Conductivity of the Lubricating Oil Film in the Contact Area

$$\alpha_2 = k/H_{i(o)0} \tag{21}$$

where $H_{i(e)0}$ is the average center oil film thickness.

According to the analysis of Hertz, the contact state of the needle roller and the raceway under the load state is line contact. Higginson and Dowson provided a numerical solution to the linear contact elasto-hydrodynamic lubrication problem, and a method for calculating the thickness of the linear contact elasto-hydrodynamic oil film, as detailed in [20]. The formula for the dimensionless minimum oil film thickness is calculated as follows:

$$H_{min} = 2.65 \frac{G^{0.54} U^{0.7}}{W^{0.13}} \tag{22}$$

$$h_{min} = [2.65\alpha^{0.54}(\eta_0 U)^{0.7} R^{0.43} L^{0.13}]/(E'^{0.03} W^{0.13}) \tag{23}$$

The relationship between the average center oil film thickness and the minimum oil film thickness is shown as follows:

$$H_0 = 4/3 \times h_{min} \tag{24}$$

where $U$ and $W$ are the introduced intermediate variables related to speed and load; $G = \alpha_{oil} E_0$; $E_0$ is the equivalent elastic modulus; and $\alpha_{oil}$ is the lubricating oil viscosity coefficient.

### 3.3. Friction Power Consumption Solve Process

The Fortran language was used to program the bearing element interaction force and bearing friction power consumption calculation custom subroutines, and to link the rocker arm full-type needle bearing to complete the simulation analysis modules. The solution procedure of the dynamics differential equations is shown in Figure 4. The initial conditions of dynamics differential equations—namely, the relative position and the motion vector of various components—are obtained based on the initial estimated values of the bearing component's position and motion constraint. Then, the dynamic differential equation of the rocker arm full-type needle bearing is established, and the Gear stiff (GSTIFF) integer algorithm with variable steps is used to solve it. Equations (6)–(13) can be solved by the GSTIFF variable-step integration algorithm to obtain the dynamic characteristics of the bearing. To verify whether the error meets the convergence error set, it was set to $1 \times 10^{-3}$. If "yes" (see Figure 4), the process continues to the next solution after obtaining the outputs of the motion parameters. If "no" (see Figure 4), the process is to choose a smaller step value and repeat the solving until the solving error meets the convergence error. Finally, the frictional power consumption of the rocker arm full-type needle bearing is calculated by Equations (14)–(19).

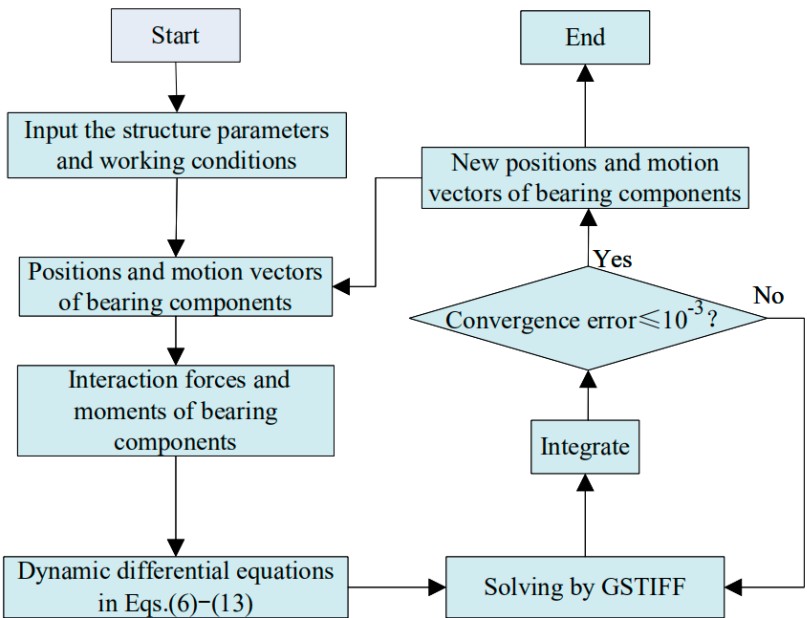

**Figure 4.** Solution procedure of dynamic of differential equation.

## 4. Simulation Analysis and Results

The bearing was selected as the simulation example, as is typical of a rock arm application, and its design parameters are shown in Table 1. The ambient temperature was 40 °C, FVA-M lubricating oil was used for lubricant, and the bearing material was GCr15.

**Table 1.** Basic parameters of the bearing.

| Parameters | Value |
| --- | --- |
| Outer diameter/mm | 24 |
| Outer ring width/mm | 6.9 |
| Central shaft width/mm | 17.3 |
| Pitch diameter/mm | 16.4 |
| Needle diameter/mm | 2 |
| Needle length/mm | 6.8 |
| Number of rollers | 26 |

*4.1. The Influence of Working Conditions on Bearing Frictional Heat Value*

In order to analyze the value of frictional heat, the outer ring initial speed was set as 7000 rpm, and the radial force was 2000 N~4500 N. The initial radial load was set as 3000 N, and the outer ring speed was 4000 rpm~9000 rpm. By solving the dynamics model, the minimum oil film thickness, the force of adjacent rollers, the half width of the contact area, and the contact force of the bottom roller can be calculated. The results obtained from the dynamic simulations are shown in Table 2. Furthermore, the partial and overall frictional heat can be extracted during the stable working of the full-type needle bearing. Figures 5 and 6 depict the variations, in average values with working conditions.

From the perspective of the primary and secondary heat sources, the order of internal heat generation is as follows: heat generated by needle rollers stirring oil, heat generated by sliding between needle rollers, and heat generated by elastic hysteresis or differential sliding. It has the same distribution law as other common heat generation models. The heat generated by the mixing of rolling elements and lubricating oil is the main part. The full-type needle bearing that only bears a radial load has load-bearing and non-loading areas, which affect the operation of the bearing and change the speed of the rolling element at this place of transition. The phenomenon generates contact collision force and friction between needle rollers, so the sliding heat generation between the rolling elements of

the full-type needle bearing follows. The heat value caused by the elastic hysteresis and differential sliding in the dynamic process is very small.

**Table 2.** The results obtained from dynamic simulations.

| Speed/r/min | Radial load/N | The Minimum Inner Oil Film Thickness/mm | The Minimum Outer Oil Film Thickness/mm | Contact Width/mm | The Contact Force/N | The Force of Adjacent Rollers/N |
|---|---|---|---|---|---|---|
| 4000 | | $2.30 \times 10^{-4}$ | $2.70 \times 10^{-4}$ | 0.0297 | 479 | 0.64 |
| 5000 | | $2.80 \times 10^{-4}$ | $3.40 \times 10^{-4}$ | 0.0297 | 479 | 0.67 |
| 6000 | 3000 | $3.10 \times 10^{-4}$ | $3.50 \times 10^{-4}$ | 0.0297 | 479 | 1.17 |
| 7000 | | $3.40 \times 10^{-4}$ | $3.90 \times 10^{-4}$ | 0.0297 | 479 | 1.42 |
| 8000 | | $3.70 \times 10^{-4}$ | $4.20 \times 10^{-4}$ | 0.0297 | 479 | 1.72 |
| 9000 | | $4.00 \times 10^{-4}$ | $4.60 \times 10^{-4}$ | 0.0297 | 479 | 2.54 |
| | 2000 | $3.70 \times 10^{-4}$ | $4.20 \times 10^{-4}$ | 0.0242 | 320 | 1.18 |
| | 2500 | $3.50 \times 10^{-4}$ | $4.00 \times 10^{-4}$ | 0.0271 | 400 | 1.47 |
| 7000 | 3000 | $3.40 \times 10^{-4}$ | $3.90 \times 10^{-4}$ | 0.0297 | 479 | 1.42 |
| | 3500 | $3.30 \times 10^{-4}$ | $3.80 \times 10^{-4}$ | 0.0322 | 638 | 1.38 |
| | 4000 | $3.30 \times 10^{-4}$ | $3.80 \times 10^{-4}$ | 0.0343 | 728 | 1.36 |
| | 4500 | $3.20 \times 10^{-4}$ | $3.70 \times 10^{-4}$ | 0.0365 | 818 | 1.52 |

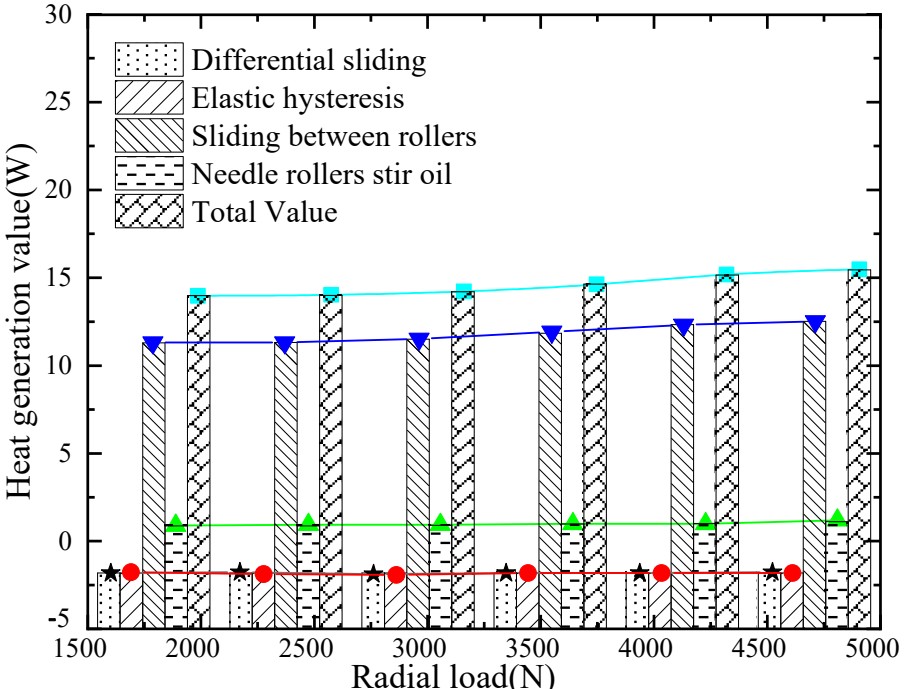

**Figure 5.** The influence of radial load on heat generation.

From the perspective of the change in heat generation with working conditions, the change in speed has a braking effect on the change in heat generation. Due to the gradual increase of the lubricating oil film thickness with the increase of the speed and the centrifugal force of the rolling elements, the value of the heat generated also gradually rises. However, the change in radial load barely affects the change in frictional heat generation. The frictional heat generation rises slightly with an increase in radial load.

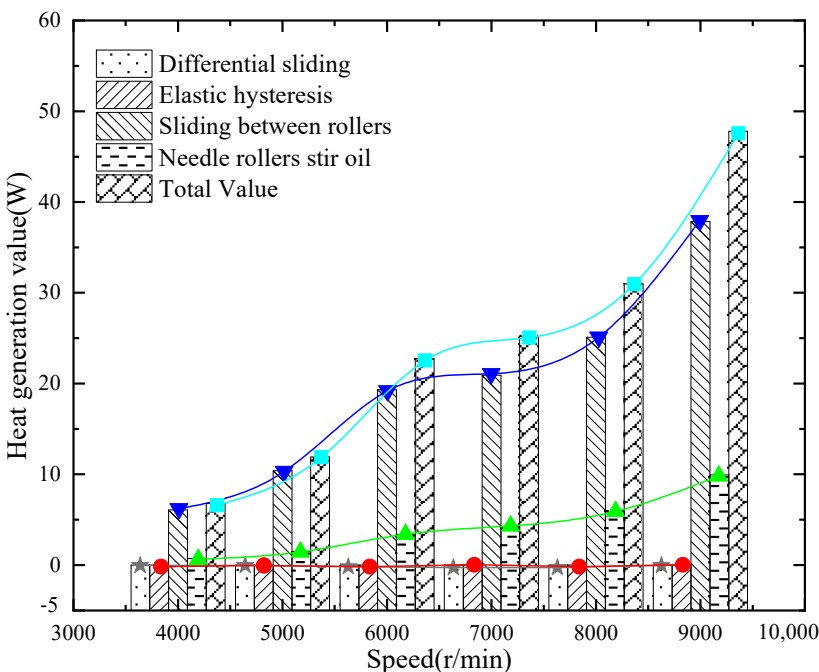

**Figure 6.** The influence of speed on heat generation.

### *4.2. Calculation of Convective Coefficient*

4.2.1. Convective Coefficient between the Bearing's Components and the Heat Transfer Medium

In the working of open bearings, it is not only necessary to consider the convection of the lubricating oil on the surface of the bearing component, but also to consider the convection between air and the surface of the component. he convective coefficient between the bearing components and the heat transfer medium at different speeds may be calculated. The results are shown in Figures 7 and 8.

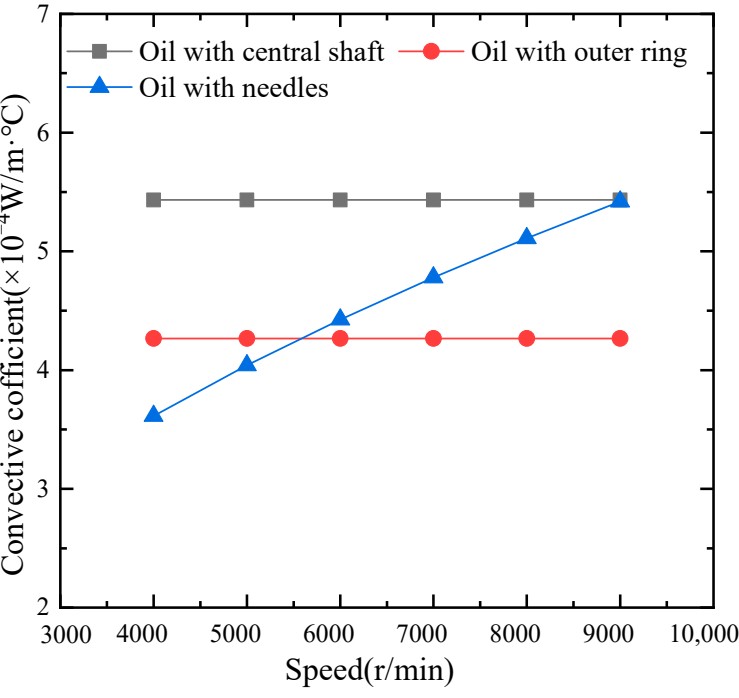

**Figure 7.** The influence of speed on the convective coefficient of oil.

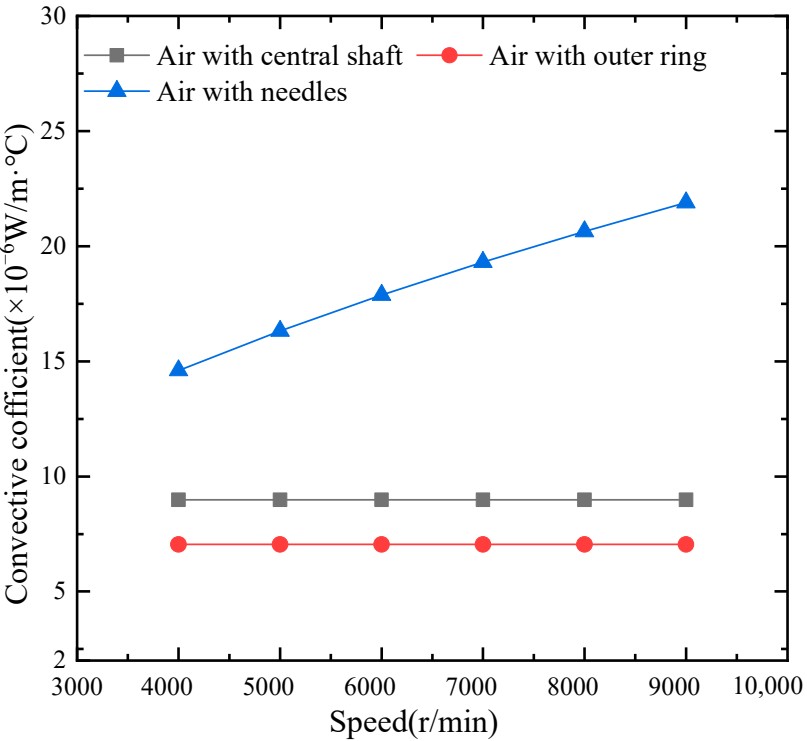

**Figure 8.** The influence of speed on the convective coefficient of air.

### 4.2.2. Thermal Conductivity Coefficient of Oil Film

During the rotation of oil-lubricated bearings, the two friction pairs between the contact areas are often separated by a layer of oil film to reduce friction and improve the bearing capacity of the bearing. Therefore, it is necessary to consider the thermal conductivity coefficient of the contact oil film between the needle roller and rings of the bearing. The changes of the thermal conductivity coefficient with the radial load and the outer ring speed are shown, respectively, in Figures 9 and 10.

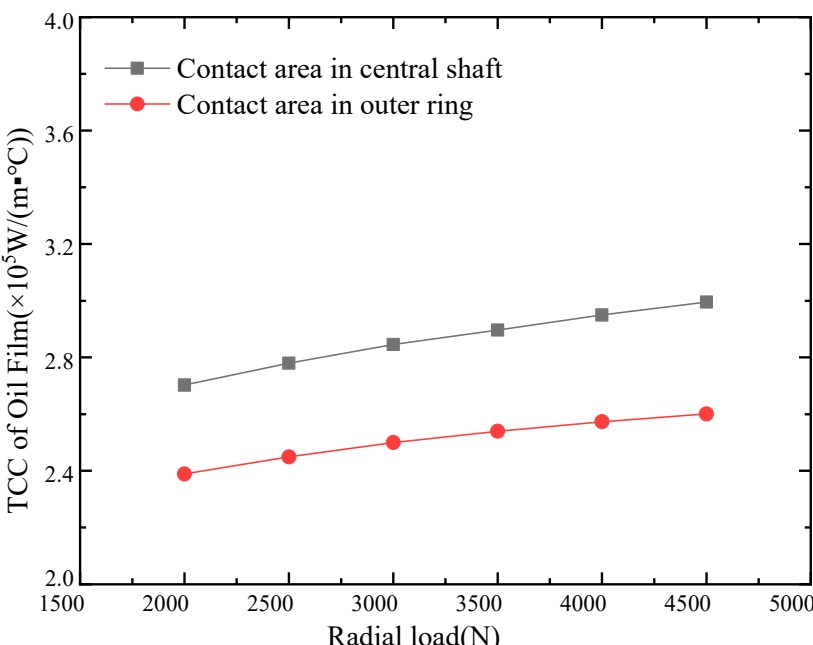

**Figure 9.** The influence of radial load on the heat transfer coefficient of oil film.

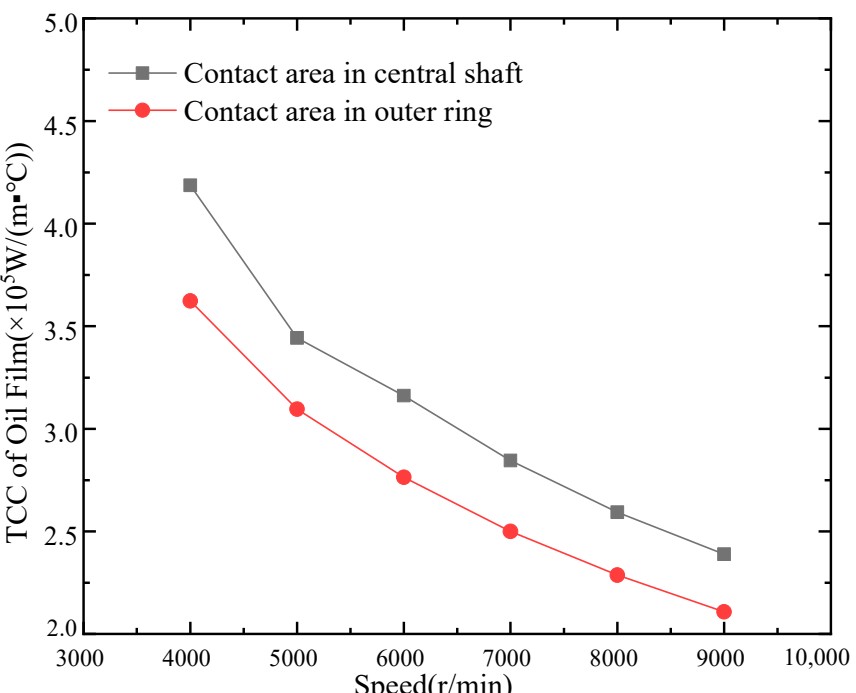

**Figure 10.** The influence of speed on the heat transfer coefficient of oil film.

From Figure 9, it can be seen that as the load increases, the convective coefficient in the contact area gradually increases. This is because when the shear force and pressure inside the oil film increase, it is difficult to form a lubricating oil film in the contact area. Therefore, it causes the thickness to gradually decrease. From Figure 10, it can be seen that as the rotation speed increases, the convective coefficient of the contact area decreases. This phenomenon is mainly caused by the lubricating oil film on the contact areas, which forms easily and gradually increases.

### 4.3. Thermal Analysis Simulation Process

The finite element method can be used to obtain the temperature field of each component of the bearing by solving the discretization area. Because the calculation model is consistent with the actual structure, the finite element method can clearly calculate the balance of each component in the cross-section and in the adjacent area. At the same time, the temperature distribution can better characterize the heat loss. The above descriptions are of two basic thermal field solving problems—namely, steady-state temperature field analysis and transient temperature field analysis. In the process of bearing service, the equilibrium temperature reached by the bearing neither generates heat nor loses heat. It is the operating temperature at which the bearing is running smoothly.

Because the bearing is a symmetrical rotating assembly, each needle roller has the same form of heat generation and heat transfer during high-speed rotation. It is assumed that the axial thermal characteristics of the system are consistent, and that the heat transfer only occurs in the radial direction [21]. In the working process, forced convection occurs between the lubricating oil and the needle rollers, as well as in the inner and outer raceway surfaces of the bearing. Natural convection occurs between the contact area of the end of the central shaft and the outside part of the shaft with the outside air. The friction heat generated by the bearing is evenly distributed to the needle roller and the ferrule, in a ratio of 1:1 [22].

#### 4.3.1. Thermal Boundary Conditions

In ANSYS Workbench, the bearing is modeled and meshed according to known parameters. The finite element model of the bearing is meshed by a mapping method. The

meshing produces a total of 114,578 elements and 525,468 nodes. The divided model is shown in Figure 11.

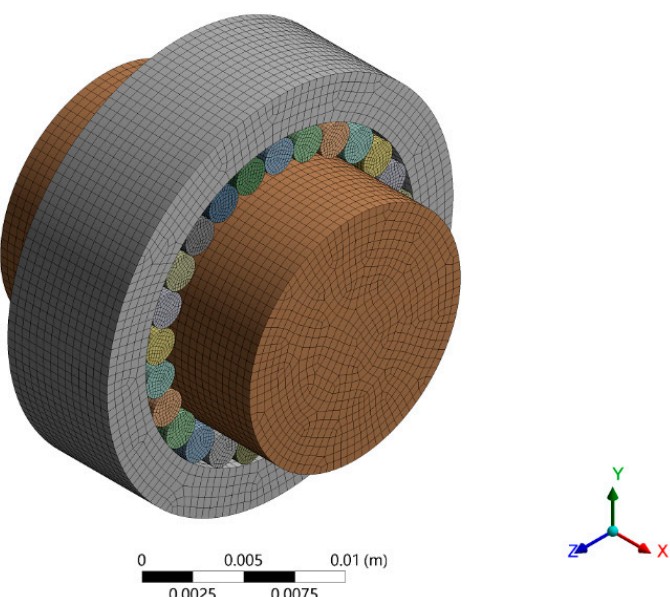

0          0.005          0.01 (m)
   0.0025        0.0075

**Figure 11.** Meshed model.

In the software, we entered the material properties, such as Young's modulus, the Poisson ratio, the density, and the thermal conductivity. We set the bearing material properties in ANSYS and added the thermal conductivity of the bearing material to 40.1 W/(m·K). Figure 12 shows the boundary conditions for the steady-state temperature field analysis of the bearing. The main settings in the steady-state temperature analysis module included three types of boundary conditions. The first type of boundary condition was setting the ambient temperature to 40 °C, i.e., the temperature of the bearing components and the external environment. The second type of boundary condition was applying a thermal load to the bearing and loading a calorific value in the form of heat flow to the inner and outer surfaces of the rolling element and the raceway in the contact area. The heat generation of the bearing was mainly on the raceway surface and the needle roller surface in the contact area. Therefore, according to the calculation results in Figures 5 and 6, the heat generation of the inner ring, the outer ring, and the needle rollers were distributed, respectively, to the three contact surfaces. The third type of boundary condition was applying the oil film convective coefficient to the contact pair and loading the convective coefficient of the air or lubricating oil on the inner and outer rings and the outer surfaces of the rolling elements. Among them, the convective heat transfer coefficients of the outer ring, the central shaft, and the surface of the needle rollers were set according to the calculation results in Figures 7 and 8, respectively. The properties of the oil film in the thermal analysis were considered as the value of the oil film conductivity coefficient in the contact pair setup. The value was related to the oil film thickness and the thermal conductivity of the lubricating oil. This was regarded as the thermal resistance between the two contact pairs. The above boundary conditions included the heat transfer and the heat dissipation boundary conditions of the bearing.

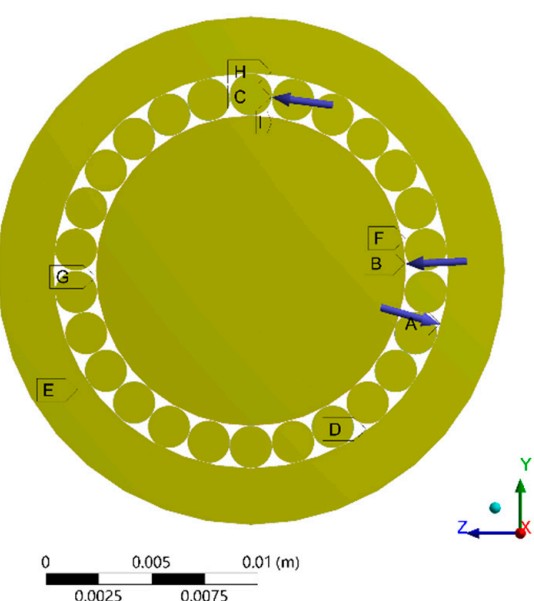

**Figure 12.** Boundary conditions for steady-state temperature field analysis.

4.3.2. Analysis of Steady-State Temperature Field

1.    The influence of oil film thermal conductivity coefficient on bearing temperature.

In order to analyze whether the convective coefficient of the lubricating oil film affects the heat dissipation inside and outside the bearing, a "path" from the center of the central axis along the positive direction of the Y axis to the outer surface of the outer ring could be set. In addition, the temperature on the "path" could be studied under the two conditions of the oil film convective coefficient and no-oil film convective coefficient. The law of change is shown in Figure 13. It can be seen from the temperature distribution that, due to the compact structure of the needle roller bearing, the temperature difference between the internal parts of the bearing was small. The temperature range of the bearing had the same distribution in each needle and contact area, and only changed in temperature along the radial direction.

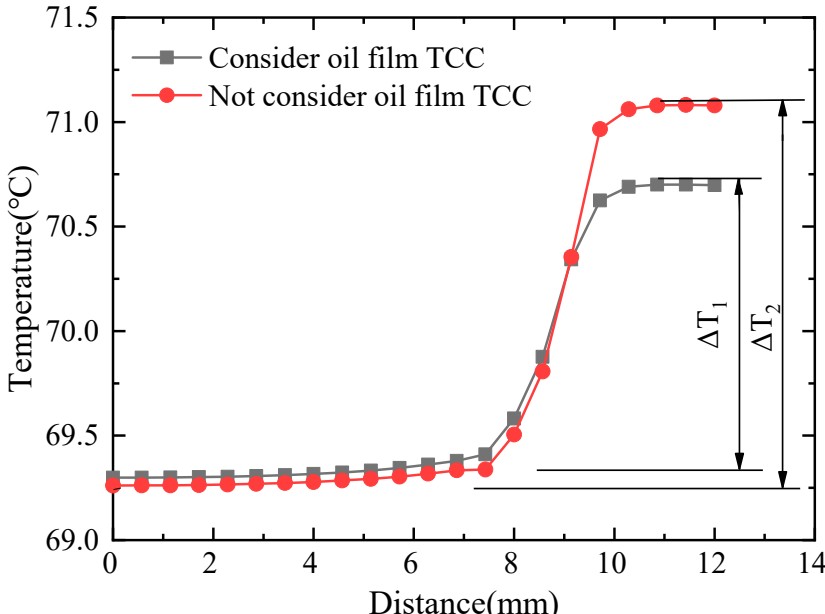

**Figure 13.** Extraction of bearing internal temperature value.

Because the outer ring rotates and the load mainly acts on the outer ring, the heat generated in the contact area between the needle roller and the outer ring is greater than the heat generated between the needle roller and the central shaft in the dynamic simulation. Accordingly, in the temperature field analysis, it can be seen that the temperature from the central shaft core to the outer ring rose and then decreased gradually, and the temperature was highest at the contact point between the needle rollers and the outer ring. In addition, when there was no heat transfer coefficient of the lubricating oil film, it meant that the bearing needles were in direct contact with the outer ring and the central shaft. As a result, the heat conduction effect was better, and the bearing temperature difference was smaller. However, when there was a lubricating oil film heat transfer coefficient, the temperature difference between the inside and outside of the bearing was large. The above phenomenon indicated that the lubricating oil films affected the heat conduction in the bearing contact area.

2. The influence of operating parameters on bearing temperature.

Using the local heat generation model, the frictional heat generation, and the convective coefficient obtained in Section 4.1 as the boundary conditions for the temperature field calculation, the bearing temperature under different loads and different speeds could be calculated. The maximum temperature of the outer ring raceway, the needle roller, and the central shaft surface could be extracted.

Figure 14 shows the effect of different radial loads on the maximum temperature when the outer ring rotated at a speed of 7000 rpm and an initial temperature of 40 °C. It can be seen that the maximum temperature rise was in the outer ring raceway. The maximum temperature value gradually decreased from the outer ring raceway, the outer surface of the outer ring, and the needle roller surface to the central shaft surface. The radial load had no effect on the heat generation of the needle roller bearing. Therefore, under the radial load of 2000 N~4500 N, the temperature of the bearing gradually increased, and the maximum temperature value was between 65 °C and 70 °C.

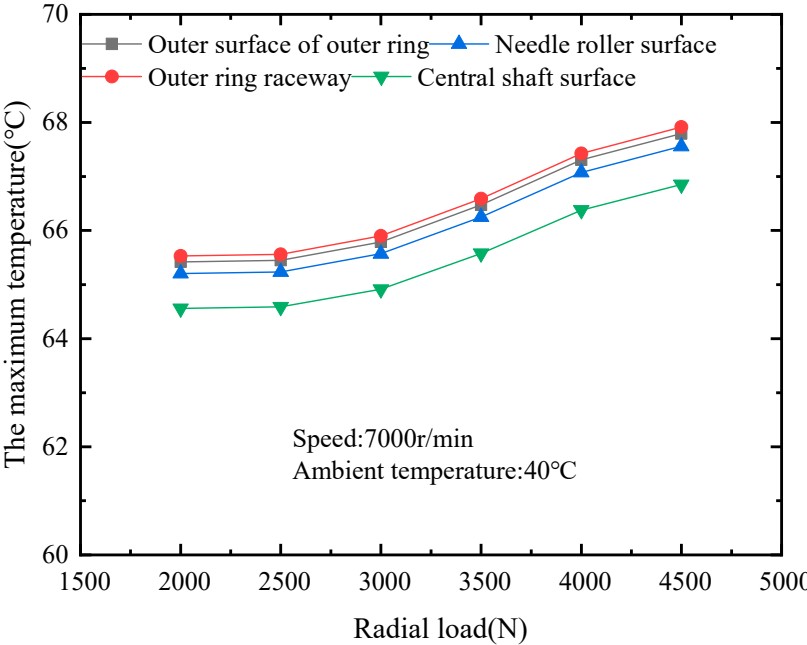

**Figure 14.** The influence of radial load on the maximum temperature of the bearing.

Figure 15 shows the influence of different speeds on the maximum temperature at a radial load of 3000 N and an initial temperature of 40 °C. It can be seen that the maximum temperature rise was also in the outer ring raceway. The maximum temperature value gradually decreased from the outer ring raceway, the outer ring outer surface, and the

needle roller surface to the central shaft surface temperature. When the speed was 9000 rpm, the maximum bearing temperature was 85 °C.

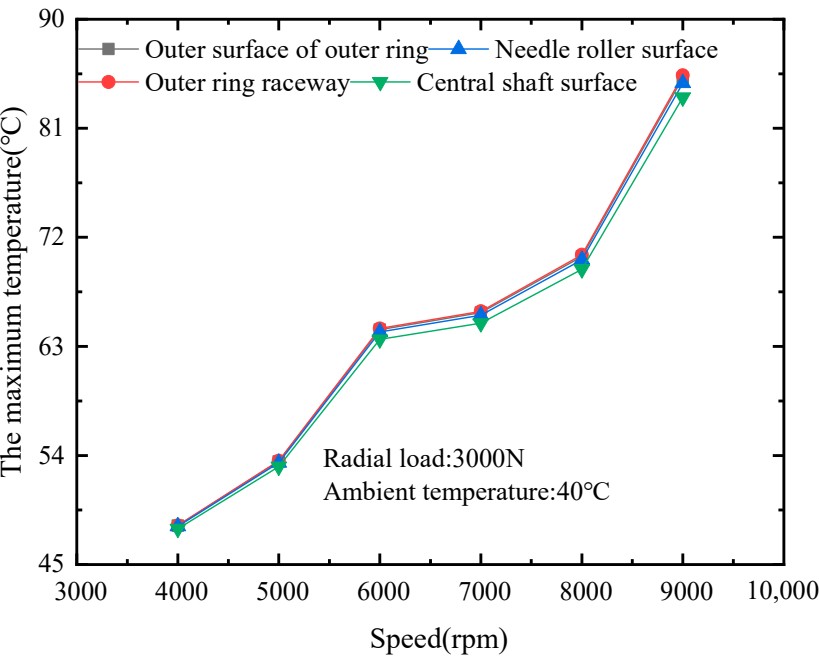

**Figure 15.** The influence of speed on the maximum temperature of the bearing.

## 5. Discussion of Simulation Results

In order to verify the correctness of the simulation method in the study, its results were calculated and compared based on the reliability test conditions of a full-type needle bearing for rocker arms of the same specifications. The rated dynamic load of the bearing was calculated as 8.6 KN. The limit speed was referred to in the SKF bearing catalog (sample) for the needle roller bearings of the same specification. The test lubricating oil was engine oil 10W-30, and its performance was similar to that of the FVA-M lubricating oil. The test conditions that were determined are shown in Table 3 [23].

**Table 3.** Test condition for bearing.

| Parameters | Value |
|---|---|
| Load/N | 2580 |
| Rotational speed of outer ring/rpm | 7000 |
| Lubricant | engine oil 10W-30 |
| Oil temperature/°C | 40 |
| Ambient temperature/°C | 40 |

The following points can be observed from Figure 16. The highest temperature distribution was on the raceway. Similarly, the raceway surface was the most prevalent failure site due to excessive temperature, which was consistent with the results of [10]. Considering the influence of load and speed, the maximum temperature of this bearing under the selected FVA-M lubricating oil was 65.9 °C, and the maximum temperature rise was 25.9 °C. That proved the rationality of the bearing conforming to the tribological design. On the one hand, the working load used in the simulation analysis was 0.2~0.3 times the rated dynamic load, and the working speed was 0.2~0.6 times the limit speed within the rated thermal speed range of the bearing specified in the fatigue life test. As a result, the temperature variation range was in line with the national standards of bearing life and the reliability test and evaluation [24]. In other words, the bearing temperature rose generally and did not exceed 55 °C above the ambient temperature, and the maximum temperature

of the outer ring of the bearing did not exceed 95 °C when lubricated by circulating oil. On the other hand, the same thermal boundary conditions as those in [15] were used in the thermal analysis of the example bearing. It is believed that when the bearing rotates at a high speed, the temperature of all parts of the raceway is basically uniform. When the bearing was modeled, the temperature coupling of the inner and outer ring surfaces, respectively was performed. It was proved by the related temperature field experiment that the simulation results were basically in agreement with the experimental data.

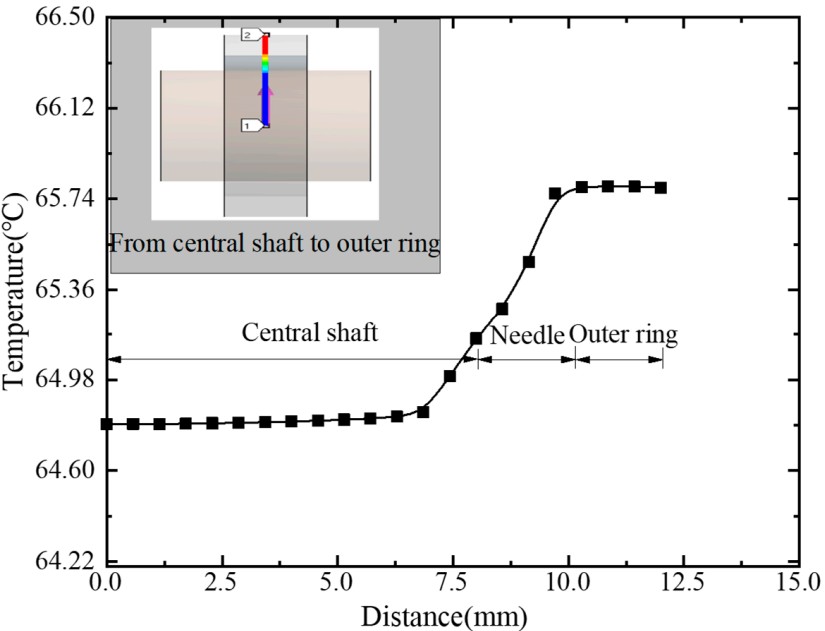

**Figure 16.** Temperature distribution under test conditions.

Thermal analysis using the finite element method can significantly enhance calculation efficiency and repeatability. In future work, we can design a thermal study of the ANSYS and Adams co-simulation based on the bearing model acquired in the secondary-built Adams commercial program. MATLAB was used to process the steady friction power consumption and convection coefficient, and the ANSYS parameterized command flow was invoked to complete the thermal analysis of the bearing. This allowed for the optimization of bearing structure characteristics, as well as for the analysis of working conditions. The method developed for analyzing bearing thermal behavior can be applied to the examination of comparable bearings. This research is extremely important for tribological design and bearing application choices.

## 6. Conclusions

Combined with the dynamic method of rolling bearings, the friction power consumption of full-type needle bearings for rocker arms under FVA-M oil lubrication is clarified. This work established a dynamic model, and the influence of the working conditions on the bearing temperature field, from the perspective of theoretical analysis and finite element calculation, was analyzed. The main conclusions are as follows:

1. The multi-body dynamics method was used to obtain the dynamic performance of bearing, and the influence of friction power consumption on the bearing temperature was studied. The results showed that as the speed increases, the frictional power consumption of the bearing increases, and the maximum temperature of the bearing gradually increases. With the increase in the load, the frictional power consumption gradually increased, and the maximum temperature of the bearing also gradually increased. Compared with the speed, the load change had a relatively small effect on the temperature of the bearing.

2. We selected appropriate boundary conditions in the bearing thermal model to study the steady-state temperature field. The results showed that the temperature of the outer raceway is the maximum, and the temperatures of the needle roller surface, the outer ring surface, the inner raceway, and the central shaft core decreased, in order. The temperature difference between the inside and outside of a bearing was affected by the lubricating oil film in the contact area of the bearing. The presence or absence of the oil film can affect the temperature difference within and between the bearing and its external environment.

3. According to an analysis of the heat generation characteristics of the bearing under the influence of multiple factors, the heat value created by the collision and sliding in a full-type needle bearing accounted for a considerable share. As a result, effective lubrication and timely lubricant replenishment should be employed in bearing applications. Under a high-speed condition, the influence of bearing thermal effects must also be considered. The temperature distribution of the bearing showed regularity. In addition, the highest temperature was distributed at the raceway. The high temperature rise reduced the hardness of the material, resulting in a contact fatigue failure between the needle roller and the raceway. To avoid that, the characteristics of the bearing material should be improved by controlling the heat treatment.

**Author Contributions:** Conceptualization, G.Z., B.S. and F.L.; data curation, G.Z. and F.L.; formal analysis, G.Z. and B.S.; funding acquisition, W.Z.; investigation, G.Z., F.L. and B.S.; methodology, B.S. and W.Z.; project administration, B.S. and H.Y.; resources, H.Y.; software, G.Z.; supervision, W.Z. and B.S.; validation, W.Z. and F.L.; writing—original draft, G.Z. and B.S.; writing—review & editing, B.S., W.Z., and H.Y. All authors have read and agreed to the published version of the manuscript.

**Funding:** This research was funded by Youth Program of the National Natural Science Foundation of China (51905152).

**Institutional Review Board Statement:** Not applicable.

**Informed Consent Statement:** Not applicable.

**Data Availability Statement:** The data used to support the findings of this study are available from the corresponding author upon request.

**Acknowledgments:** The authors would like to thank Youth Program of the National Natural Science Foundation of China (51905152) for the financial support.

**Conflicts of Interest:** The authors declare no conflict of interest.

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
