# Peer review of "Thermal Analysis Based on Dynamic Performance of Rocker Arm Full-Type Needle Bearings"

_lubricants, doi:10.3390/lubricants10050104_

Round 1

Reviewer 1 Report

Questions and remarks.

  1. Whether lubricating oil film is present in the contact area affects the heat conduction of the bearing, which results in a temperature difference. However, the authors do not indicate how the property of the lubricant to be adsorbed on surfaces was taken into account.
  2. (line 221) μjmi, μjmo are respectively the traction coefficients of the oil film between the mth slice of the jth needle and the inner and outer raceways, which are obtained by interpolation from the relationship between the slip-roll ratio and the elasto-hydrodynamic tration coefficient. The values of these coefficients are not presented in the initial data.
  3. (line 375) The finite element method can be used to obtain the temperature field of each component of the bearing by solving the discretization area. The meshing produces a total of 318,448 elements and 459,000 nodes. The thickness of the lubricating film is very thin. How many finite elements were considered along the film thickness to take into account its properties?
  4. (line 450) The test conditions determined are shown in Table 2. Oil temperature/°C =100. Is this the oil supply temperature? Or what? In Figure 16. (Temperature distribution under test conditions) the calculated temperature level is below 100.
  5. In the text (line 342) the term outer ring is used in the form "outer ring". Figures 7 and 8 use "outerring". Figures 9 and 10 use "outrring".

Reviewer 2 Report

Dear authors,

As for the content page, the work is acceptable.

My recommendations:

  • In chapter "4. Simulation analysis and results", improve pictures quality (Fig. 5, 6), „Differential sliding“ and „Elastic hysteresis“ are on the same level.
  • Highlight the results obtained from dynamic simulations (eg summary table).
  • Describe the inputs for thermal analysis in Ansys in more detail, consider finer mapped mesh for model, model is suitable for mapped mesh
  • Complete the chapter "4 Discussion" about a description of your solution's scientific and practical contribution. The findings and their implications should be discussed in the broadest context possible. Future research directions may also be highlighted.
  • The conclusion could be extended about the possibility of a next specific application of the proposed solution in practice.
  • In the manuscript, please emphasize the novelty of your approach.

Overall, however, the article is interesting. I wish you good luck in further research.

Kind regards

Reviewer
